# Macronutrients Intake and Risk of Stomach Cancer: Findings from Case-Control Study

**DOI:** 10.3390/nu14122373

**Published:** 2022-06-08

**Authors:** Sabika Allehdan, Maya Bassil, Razan M. Alatrash, Tareq Al-Jaberi, Ahmad Hushki, Yaser Rayyan, Mahammad Dahoud, Khaled Abu-EIteen, Reema F. Tayyem

**Affiliations:** 1Department of Biology, College of Science, University of Bahrain, Sakhir Campus, Zallaq 32038, Bahrain; sallehdan@uob.edu.bh; 2Department of Human Nutrition, College of Health Sciences, QU Health, Qatar University, Doha P.O. Box 2713, Qatar; bassil.maya@qu.edu.qa; 3Department of Health Education, Jordan University Hospital, Amman 11942, Jordan; reta_94_atrash@hotmail.com; 4Department of General & Pediatric Surgery, Faculty of Medicine, Jordan University of Science and Technology, Irbid 22110, Jordan; tmrjaberi@hotmail.com; 5Gastroenterology Division, King Hussein Cancer Center, Amman 11941, Jordan; ah.11233@khcc.jo; 6Department of Gastroenterology & Hepatology, Faculty of Medicine, The University of Jordan, Amman 11942, Jordan; rayyan@marshli.edu; 7Nursing Department, King Hussein Cancer Center, Amman 11941, Jordan; betawi114@yahoo.com; 8Department of Biology, Faculty of Science, The Hashemite University, Zarqa 13133, Jordan; salma@hu.edu.jo; 9Department of Nutrition and Food Technology, Faculty of Agriculture, The University of Jordan, Amman 11942, Jordan

**Keywords:** case-control study, gastric cancer, macronutrients, fat

## Abstract

Studies on the association between gastric cancer (GC) and the intake of nutrients in Jordan are very limited, while findings from other reports on the intake of energy and macronutrients are controversial. This study aimed to examine the associations between intake of energy and macronutrients and the risk of GC in a Jordanian population. A case-control study was carried out between March 2015 and August 2018 in four major hospitals, including an oncology center in Jordan. Study participants were 173 cases with incident and histologically confirmed GC and 314 frequency-matched controls. Interview-based questionnaires were used to obtain the study’s information. Data on nutrient intake were collected using a validated Arabic food-frequency questionnaire (FFQ). Odds ratios (ORs) and their corresponding 95% confidence intervals (CIs) were calculated through multinomial logistic regression and adjusted for potential confounders, including age, marital status, education, body mass index (BMI), smoking, period of smoking, family history of gastric cancer, history of gastric ulcer, and physical activity. Intakes of total fat, saturated fat, monounsaturated fat, polyunsaturated fat, cholesterol, trans-fat, and omega-6 fatty acids were significantly associated with increased risk of GC. The ORs for the highest versus the lowest tertiles were 6.47 (95% Cl: 3.29–12.77), 2.97 (95% CI: 1.58–5.58), 6.84 (95% CI: 3.46–13.52), 6.19 (95% CI: 3.15–12.17), 3.05 (95% CI: 1.58–5.88), 8.11 (95% CI: 4.20–15.69), and 2.74 (95% CI: 1.47–5.09), respectively. No significant association was found for energy, protein, carbohydrate, sugar, fibers, and *omega*-3 fatty acids. The findings of this study suggest that high intake of selected types of fats was associated with an increased risk of GC.

## 1. Introduction

Gastric cancer (GC) is one of the most common malignant cancers in the world. Despite the fact that its global incidence rate is declining over the last decades [1], GC is considered the fifth most common diagnosed cancer and the third leading cause of cancer death in both genders worldwide [2]. More than 70% of GC cases are in developing countries such as Asia, South America, and Eastern Europe [3]. GC incidence is low in Jordan, contributing to 2.8% of all newly diagnosed cancer cases. It affects men more than women, with a ratio of 1.7:1. GC mortality accounts for 3.5% of all oncological deaths, ranking eighth among the top 10 cancer-related mortality causes in Jordan [4].

Several risk factors are associated with the development of GC, including genetic factors [5], *Helicobacter pylori* infection [5,6,7,8], tobacco smoking [9], heavy alcohol drinking [10,11], and obesity [12,13]. Moreover, dietary habits and intake of nutrients are considered to have an important association with GC risk [14]. Numerous studies have examined the associations between dietary factors and GC and report conflicting findings. Evidence from some studies found a positive association between risk of GC and protein intake [15,16,17], fat [18,19,20], saturated fat [17,20,21,22], cholesterol [19,20], and carbohydrates [23,24]. However, other studies reported negative or no associations with protein, monounsaturated, polyunsaturated fats [25,26], and carbohydrates [27,28,29,30]. Findings were inconsistent for fiber intake; some studies revealed negative associations with the risk of GC [20,31], whereas other studies found no association [28,32,33].

In summary, and in view of these conflicting results [17,20,34,35,36], we aimed in the present study to give further understanding into the association between stomach cancer risk and the intake of macronutrients in a case-control study conducted in Jordan, where dietary intake was assessed using a reliable and valid food frequency questionnaire (FFQ).

## 2. Methods 

### 2.1. Study Design and Participants

This case-control study was carried out between March 2015 and August 2018 in Jordan. Two hundred and nine GC patients with an incident and histologically confirmed GC were invited to participate in this study. However, 173 GC patients with either gastric adenocarcinoma (85%) or other types of GC (gastrointestinal stromal tumor, mucosa-associated lymphoid tissue (MALT) lymphomas, and carcinoid tumor) (15%) accepted participation in this study. Majority of GC cases were in stages three (28.9%) and four (57.8%). The control group comprised of 314 participants (hospital workers, patient visitors, patient escorts, university students, and university workers) was recruited conveniently from community. Individuals who did not have a previous history of GC or any type of cancer were selected and frequency matched to cases by age, sex, employment status, and marital status.

Jordanian participants aged 18 years or older, able to communicate verbally, and free of any chronic diseases that need dietary interventions were included in this study. Exclusion criteria included being critically ill, hospitalized, and unable to communicate verbally; pregnant or breastfeeding women; as well as being diagnosed with cancer for more than six months or being affected by GC as a second cancer. A signed written informed consent was obtained from each participant before their enrollment in the current study.

Cases and some controls were enrolled from four major hospitals with an oncology center service in Jordan. These hospitals were King Hussein Cancer Center, King Abdullah University Hospital, Jordan University Hospital, and Al-Bashir Hospital. The study protocol followed the ethical guidelines of the 1975 Declaration of Helsinki and was approved by the Institutional Review Board Ethics Committee of the King Hussein Cancer Center (IRB No. 15 KHCC 03, Amman, Jordan), King Abdullah University Hospital, Jordan University Hospital, and Al-Bashir Hospital.

### 2.2. Data Collection

A structured questionnaire was used for data collection from both cases and controls by trained nutritionists. This structured questionnaire included data on socio-demographic characteristics, such as age, gender, marital status, education, employment status, anthropometric measurements, family income/month, tobacco smoking, family history of GC, and history of stomachache and stomach ulcers.

### 2.3. Anthropometric Assessment

The participants’ body weight and height were measured using standardized scales by trained nutritionists. Body weight was measured to the nearest 0.1 kg using digital scale (Seca, Germany), with minimal clothing and without shoes. Height was measured to the nearest cm with the participant standing without shoes using stadiometer (Seca, Germany). Both cases and controls were asked to recall their pervious weights. Previous and current body mass index (BMI) were computed by dividing weight in kilograms by the square of height in meters [37].

### 2.4. Physical Activity Questionnaire

The physical activity level of both cases and controls was estimated using a 7-day physical Activity Recall (PAR) questionnaire. The 7-day PAR is an organized and validated questionnaire that depends on the participant’s recall of number of hours spent on doing physical activity over a week [38]. The weekly physical activity frequency, intensity, period, and type were obtained to assess the level of physical activity. All physical activities even sleeping were allocated an intensity level into metabolic equivalents (MET) (sleeping = 1.0 MET, light activity = 1.5 METs, moderate activity = 4.0 METs, and vigorous ≥ 7 METs) [39]. Physical activity was computed as the time (min) each activity was performed multiplied by the corresponding MET for that activity and multiplied again by the number of days that the activity was done per week, as expressed in the following equation: (MET level × min of activity/day × days per week) [38].

### 2.5. Dietary Assessment

A validated Arabic quantitative FFQ was used to assess dietary intake over last 12 months [40]. The FFQ questions examined the data dealing with the participant’s dietary history. A trained nutritionist asked participants to recall how frequently they had eaten or drank one standard serving of specific food items during the last year. The portion sizes of each food item were defined according to frequently used portion sizes, for example, standard measuring tools (e.g., cups, tablespoons, and teaspoons) and natural units or typical units for some foods. Food models and standard measuring units were used to assist participants in estimating the consumed portion size of foods correctly. Participants’ answers were transformed into average daily intake (in grams for foods or milliliters for drinks) based on the frequency and portion size of a specific food item. Dietary intake was analyzed by dietary analysis software (Food Processor SQL version 10.1.129; ESHA, Salem, OR, USA), which modified with additional information on popular Jordanian cuisine to estimate daily intakes of energy and macronutrients [41].

### 2.6. Statistical Analysis

Statistical analysis was conducted using the Statistical Package for the Social Sciences (SPSS) version 27 (IBM Corporation, Armonk, NY, USA). A *p*-value < 0.05 was considered significant. Data are displayed as mean ± standard error of mean (SEM) for normally distributed continuous variables such as age, anthropometric measurements, monthly income, and physical activity level. Independent sample *t*-test was used to find differences in normally distributed continuous variables between GC cases and controls. Frequencies and percentages were calculated to present categorical data. Pearson’s chi-square was used to observe the differences in sociodemographic characteristics such as gender, marital status, education, and occupation and health characteristics such as using tobacco, family history of GC, and history of stomachache and stomach ulcer between GC cases and controls.

Because most of the nutrients are associated with total energy intake, an energy adjustment method was performed for the intake of macronutrients using the residual method of Willett [42], in which residuals were calculated from a regression model. Normality of the distribution of crude and energy adjusted nutrients variables was examined by the Shapiro–Wilk test. The median and 33th and 67th percentiles were calculated for intakes of energy, crude macronutrients, and energy adjusted macronutrients. *Mann*–*Whitney U* test was used to detect differences in intakes of energy and macronutrients between GC group and control group.

Intakes of energy adjusted macronutrients were modeled using tertiles of distribution in the study population, with the first tertile being the lowest intake and third tertile the highest. Odds ratios (ORs) and 95% confidence intervals (95% CIs) for GC were calculated using a multinomial logistic regression mode according to tertile of energy and energy adjusted macronutrient intakes, with the lowest tertile as the reference group. Potential confounders (age, gender, marital status, education, BMI, smoking, period of smoking, family history of gastric cancer, presence of gastric ulcer, and physical activity (MET-min/week)) were selected based on reported risk factors for GC in previous studies [36,43,44]. *p*-value for trend was calculated using linear logistic regression model.

## 3. Results

Table 1 shows socio-demographic and health characteristics of 173 GC cases and 314 controls. No statistical differences were observed between cases and controls for age, height, monthly income, marital status, educational level, occupational status, and tobacco use. GC cases were significantly less active than controls (2203.6 ± 146.7 vs. 3356.5 ± 178.2, *p* < 0.001). The total years of smoking was longer among cases (12.4 ± 1.3) than controls (8.8 ± 0.84) (*p*-value = 0.018) Previous weight and BMI were significantly higher in GC cases compared to controls, whereas current weight and BMI were significantly lower in the case group. The proportion of participants with a family history (beyond the second degree) of GC was significantly higher in GC cases (9.8%) compared to the controls (1.6%) (*p* < 0.001). Cases had higher proportion (*p* < 0.001) of participants with gastric ulcers (43.9%) and gastric ache (27.7%) than the control group (1.6% and 4.5%, respectively).

The median daily intake and the 33rd and 67th percentile of total energy, crude macronutrients, and energy adjusted macronutrients are presented in Table 2. The GC group reported significantly higher intakes of total energy, protein, carbohydrates, fibers, sugars, fat, saturated fat, monounsaturated fat, polyunsaturated fat, cholesterol, trans-fat, omaga-3, and omega-6 fatty acids (*p* < 0.01) compared to the control group. Likewise, the GC cases had significantly higher intakes of energy adjusted macronutrients (*p* < 0.05) when compared to the control group except for protein and omega-3 fatty acids.

Table 3 presents the ORs and the corresponding 95% CIs of energy, protein, carbohydrates, fibers, sugars, various types of fats, and cholesterol. Total fat intake was positively associated with GC (OR 6.47, 95% CI: 3.29–12.77) for the highest versus the lowest tertile; *p* for trend < 0.001. The OR for saturated fat was 2.97 (95% CI: 1.58–5.58) for the highest versus the lowest tertile; *p* for trend < 0.0001. Intakes of monounsaturated fat, polyunsaturated fat, trans-fat, cholesterol, and omega-6 fatty acids were also positively associated with GC for the highest versus the lowest tertiles, with ORs of 6.84 (95% CI: 3.46–13.52), 6.19 (95% CI: 3.15–12.17), 3.05 (95% CI: 1.58–5.88), 8.11 (95% CI: 4.20–15.69), and 2.74 (95% CI: 1.47–5.09), respectively. There was no association with the intake of total energy, protein, total carbohydrates, fibers, sugars, and omega-3 fatty acids.

## 4. Discussion

The present study investigated the associations between energy and macronutrients intake and the risk of gastric cancer (GC) in a Jordanian population. Findings revealed that intakes of total, saturated, trans, monounsaturated, and polyunsaturated fats as well as omega-6 fatty acids and cholesterol were significantly and positively associated with GC risk after correcting for potential confounders (age, gender, marital status, education, BMI, smoking, period of smoking, family history of gastric cancer, presence of gastric ulcer, and physical activity).

In line with our results, high-fat diets have been consistently found to increase the risk of GC [18,19,20,26,45], which might be due to their obesogenic effects and the resulting inflammatory response [46]. Indeed, GC cases in the present study were obese prior to diagnosis and had significantly higher pre-diagnosis weight compared to controls. After GC diagnosis, weight and BMI of participants in the case group decreased despite a higher energy intake and lower physical activity, which reflects the metabolic stress associated with cancer and resulting increased energy expenditure [47]. Higher fat consumption in the case group in the present study could also be related to increased intake of saturated fats that constituted 27% of total fats. Saturated fats are known to be strong predictors of GC [17,20,21,22,36,45] as observed in our study and were found to stimulate tumorigenesis at the molecular level [48].

Similarly, the positive and significant association between increased cholesterol intake and GC risk in the present study have been previously reported [19,20]. Although the mechanisms are still not established, high cholesterol intake causes impairments in apo-lipoproteins and lipids, which may lead to inflammation [49]. As for trans-fats, higher intake was also associated with increased GC risk. Limited and inconsistent results exist in the literature on the role of trans-fats in cancer risk [50] although associations with increased risk of gastrointestinal (GI) cancer have been reported [51]. Consistently, Elaidic acid, a trans-fatty acid, was found to enhance the metastasis of GI cancer cells [52].

Elevated intakes of polyunsaturated fats, specifically omega-6 fatty acids, increased GC risk in the present study. Previous findings on the effects of polyunsaturated fats on GC risk are contradictory, whereby positive [48], negative [25,45], and no associations [36] were observed. Discrepancies could be related to the source of omega-6 fatty acids and whether it is heat-treated. In the Middle East, including Jordan [53], omega-6 fatty acids consumption is mainly derived from the use of plant oils in cooking, and thus, high temperature exposure might lead to the production of carcinogenic compounds such as polycyclic aromatic hydrocarbons [54]. In addition, increased intake of omega-6 rather than omega-3 fatty acids, as found in the present study, is associated with increased inflammation [55] and may enhance the development of cancer [48].

Additionally, our results showed that increased consumption of monounsaturated fats was positively associated with GC risk. Few studies in the literature have examined the link between intakes of monounsaturated fats and GC, and thus, far findings are inconsistent, revealing either positive [25], negative [48], or no association [36,45]. The study conducted in Mexico by Lo’pez-Carrillo et al. [25] found a positive association: monounsaturated fat intake was mainly derived from poultry, meat, and milk, all of which are foods from animal sources known to increase the risk of GC [16]. However, in the present study, the main source of monounsaturated fats in the Jordanian diet is olive oil [53], the intake of which was found to be protective against GI cancer in a previous meta-analysis [56]. This disagreement may be attributed to the different uses of olive oil in different cultures, whereby it is commonly added in cooking in the Jordanian diet and thus exposed to high temperatures, possibly leading to the production of carcinogenic compounds such as low molecular weight aldehydes [57]. Furthermore, at the molecular level, oleic acid was found to promote tumor progression and metastasis [58,59].

This is the first report that studied the associations between the intakes of energy and macronutrients and GC risk in the Arab world. Strengths of the study include the use of a validated FFQ and adjustments for many potential confounders reported in the literature. Limitations are inherent in the retrospective design of the study that prevents establishing causal relationships although this was mitigated by recruiting newly diagnosed patients with GC. Furthermore, even after adjusting for multiple factors, associations could have been affected by residual confounders and recall bias. There was no clinical confirmation for enrolled controls that were free of GC, which is one of the limitations of the current study. Moreover, the enrolled cases are a mixture of GC individuals of different etiology and histology, with majority of them presenting with gastric adenocarcinoma.

In conclusion, we found that increased total fat intake is associated with increased GC cancer among Jordanians. Higher consumption of saturated fat, monounsaturated fat, polyunsaturated fat, cholesterol, omega-6 fatty acids, and trans-fat are positively associated with GC risk. Prospective long-term studies are needed to confirm and further explore the relationships between energy and nutrient intake and GC risk. Findings can help health authorities in setting dietary guidelines for the prevention of gastric cancer in Jordan.

## Figures and Tables

**Table 1 nutrients-14-02373-t001:** Socio-demographic characteristics of the study population.

Variable	Cases (*n* = 173)	Controls (*n* = 314)	*p*-Value *
	Mean ± SEM	Mean ± SEM	
**Age (years)**	54.1 ± 0.96	54.0 ± 0.70	0.916
**Previous Weight (kg)**	85.3 ± 1.6	79.4 ± 1.2	0.003
**Current Weight (kg)**	70.6 ± 1.3	80.9 ± 0.94	<0.001
**Height (cm)**	167.9 ± 0.49	168.0 ± 0.41	0.928
**Previous BMI (kg/m^2^)**	30.1 ± 0.5	28.7 ± 0.53	0.008
**Current BMI (kg/m^2^)**	25.0 ± 0.49	28.7 ± 0.33	<0.001
**Monthly Income (JD)**	674.0 ± 36.8	575.9 ± 36.0	0.201
**Physical Activity (MET-min/week) ****	2203.6 ± 146.7	3356.5 ± 178.2	<0.001
**Period of smoking (years)**	12.4 ± 1.3	8.8 ± 0.84	0.01
**Marital Status**			
Married	148 (85.5%)	273(86.9%)	0.346
Single	8 (4.6%)	20 (6.4%)	
Divorced	3 (1.7%)	7 (2.2%)	
Widowed	14 (8.1%)	14 (4.5%)	
**Educational Level**			
Illiterate	10 (5.8%)	18 (5.7%)	0.663
Primary school	54 (31.2%)	80 (25.5%)	
high school diploma	44 (25.4%)	72 (22.9%)	
College diploma	25 (14.5%)	57 (18.2%)	
Bachelor’s degree	34 (19.7%)	71 (22.6%)	
Master’s degree	4 (2.3%)	13 (4.1%)	
Doctorate	2 (1.2%)	3 (1.0%)	
**Occupational**			
Employee	82 (59.7%)	153 (31.7%)	0.488
Non-employee	90 (40.)	161 (68.3%)	
**Family History of Gastric Cancer**			
Yes	17 (9.8%)	5 (1.6%)	<0.001
No	156 (90.2%)	309 (98.4%)	
**Tobacco Use**			
Yes	56 (32.4%)	102 (32.5%)	0.463
No	117 (67.6%)	212 (67.5%)	
**Presence of Stomachache**			
Yes	48 (27.7%)	14 (4.5%)	<0.001
No	125 (72.3%)	300 (95.5%)	
**History of Stomach Ulcers**			
Yes	76 (43.9%)	5 (1.6%)	<0.001
No	97 (56.1%)	309 (98.4%)	

* *p*-Values were calculated by independent sample *t*-test for continuous variables and Pearson’s chi-square for categorical variables. *p*-Value < 0.05 was considered statistically significant. ** MET, metabolic equivalents-minutes/week.

**Table 2 nutrients-14-02373-t002:** Median daily intake and 33rd and 67th percentile of energy and macronutrients for 173 gastric cancer cases and 314 controls participating in a case-control study between 2015–2018.

Energy and Macronutrient	Crude				Adjusted for Energy ^a^			
	Median (33rd–67th Percentile)		Median Difference Control vs. Case	*p*-Value *	Median (33rd–67th Percentile)		Median Difference Control vs. Case	*p*-Value *
	Control	Case			Control	Case		
Energy (kcal)	2339.2 (1999.1–2608.3)	3057.0 (2690.4–3581.8)	−717.8	<0.001	-----------------	-----------------	-------------	-------
Protein (g)	80.7 (71.0–94.5)	105.3 (88.5–119.5)	−24.6	<0.001	107.6 (87.8–121.2)	109.9 (103.2115.8)	−2.3	0.097
Carbohydrates (g)	298.8 (252.5–352.5)	388.8 (323.9–455.9)	−90.0	<0.001	327.5 (312.5–349.1)	407.2 (387.1–422.8)	−79.7	<0.001
Fibers (g)	23.1 (19.4–27.4)	28.4 (23.3–31.1)	−5.3	<0.001	23.3 (19.9–27.5)	27.6 (25.3–30.6)	−4.3	<0.001
Sugars (g)	122.8 (102.0–147.7)	162.3 (132.1–195.1)	−39.5	<0.001	160.9 (138.0–180.7)	173.8 (151.5–189.4)	−12.9	0.002
Fat (g)	91.4 (77.9–104.6)	129.6 (112.8–148.8)	−38.2	<0.001	99.6 (91.3–107.2)	136.4 (132.4–143.8)	−12.9	<0.001
Saturated fat (g)	25.6 (20.9–30.0)	35.6 (29.6–43.1)	−10.0	<0.001	30.0 (27.5–32.9)	37.6 (35.7–40.7)	−36.8	<0.001
Monounsaturated fat (g)	26.8 (23.3–30.3)	37.7 (33.3–42.5)	−10.9	<0.001	28.6 (25.1–32.3)	38.6 (35.4–41.2)	−10.0	<0.001
Polyunsaturated fat (g)	13.8 (10.9–17.6)	23.7 (20.2–26.4)	−9.9	<0.001	13.9 (11.2–17.9)	23.9 (22.0–26.1)	−10.0	<0.001
Cholesterol (mg)	265.4 (198.3–325.2)	333.4 (265.1–407.4)	−68.0	<0.001	276.2 (238.6–323.1)	354.5 (293.8–418.3)	−78.3	<0.001
Trans-fat (g)	0.14 (0.07–0.24)	0.41 (0.27–0.66)	−0.27	<0.001	0.14 (0.07–0.24)	0.76 (0.62–1.0)	−0.62	<0.001
Omega-3 fatty acids (g)	0.70 (0.58–0.80)	0.94 (0.77–1.1)	−0.24	<0.001	1.0 (0.79–1.2)	0.94 (0.77–1.1)	0.06	0.665
Omega-6 fatty acids (g)	6.7 (5.7–8.0)	8.0 (6.7–10.2)	−1.3	<0.001	7.1 (6.1–8.1)	8.7 (7.7–9.9)	−1.6	<0.001

^a^ Nutrient intake adjusted for total energy intake. * *p*-Values were calculated using *Mann–Whitney U* Test. *p*-value < 0.05 was considered statistically significant.

**Table 3 nutrients-14-02373-t003:** Odds ratios (ORs) and corresponding 95% confidence intervals (CIs) according to the intake of energy and selected macronutrients among 173 gastric cancer cases and 314 controls participating in a case-control study between 2015–2018.

Energy and Nutrient ^a^	OR (95%CI) ^b^			
	T1 ^c^	T2	T3	*p*-Value of Trend *
**Energy**	1	1.04 (0.57–1.90)	0.96 (0.52–1.79)	0.975
No. of cases/control	57/104	59/106	57/104	
**Protein**	1	1.35 (0.74–2.49)	1.34 (0.71–2.54)	0.235
No. of cases/control	57/104	59/105	57/105	
**Carbohydrates**	1	1.17 (0.64–2.12)	0.72 (0.38–1.35)	0.247
No. of cases/control	58/104	58/105	57/105	
**Fibers**	1	1.12 (0.61–2.05)	0.92 (0.51–1.68)	0.722
No. of cases/control	58/105	58/103	57/106	
**Sugars**	1	1.12 (0.62–2.03)	1.03 (0.57–1.89)	0.675
No. of cases/control	58/105	58/103	57/106	
**Fat**	1	**2.25 (1.11–4.57)**	**6.47 (3.29–12.77)**	**<0.001**
No. of cases/control	39/141	39/89	95/84	
**Saturated Fat**	1	1.40 (0.71–2.79)	**2.97 (1.58–5.58)**	**<0.001**
No. of cases/control	41/132	42/89	90/93	
**Monounsaturated Fat**	1	1.81 (0.93–3.53)	**6.84 (3.46–13.52)**	**<0.001**
No. of cases/control	36/144	42/89	90/93	
**Polyunsaturated Fat**	1	**2.16 (1.07–4.36)**	**6.19 (3.15–12.17)**	**<0.001**
No. of cases/control	41/133	38/92	94/89	
**Cholesterol**	1	**2.04 (1.07–3.88)**	**3.05 (1.58–5.88)**	**0.002**
No. of cases/control	39/120	60/104	74/90	
**Trans-Fat**	1	1.79 (0.82–3.93)	**8.11 (4.20–15.69)**	**<0.001**
No. of cases/control	30/147	32/77	111/90	
**Omega-3 fatty acids (g)**	1	1.3 (0.70–2.29)	0.90 (0.49–1.67)	0.999
No. of cases/control	50/114	63/101	60/99	
**Omega-6 fatty acids (g)**	1	**2.10 (1.10–4.00)**	**2.74 (1.47–5.09)**	**0.001**
No. of cases/control	44/123	50/97	79/94	

^a^ Nutrient intake adjusted for total energy intake. ^b^ Adjusted for age, gender, marital status, education, BMI, smoking, period of smoking, family history of gastric cancer, presence of gastric ulcer, and physical activity. The control group was considered the reference group for analysis. ^c^ Reference tertile. * *p*-values were calculated using linear logistic regression test. *p*-value < 0.05 was considered statistically significant.

## Data Availability

Not applicable.

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
