# Peer review of "Macronutrients Intake and Risk of Stomach Cancer: Findings from Case-Control Study"

_nutrients, 2022, doi:10.3390/nu14122373_

Round 1

Reviewer 1 Report

This is a well written case-control research study on macronutrient intake and risks of stomach cancer.  The methods, data analysis and results are clearly written and support the conclusion provided.

Author Response

Dear Reviewer 1:

We would like to thank you and the reviewers for taking the time to carefully review our manuscript and we sincerely appreciate their helpful suggestions to improve the quality of our manuscript. Following the reviewers’ concerns and comments, we made some modifications to the initial version of our manuscript which we marked by track changes. We also rewrite some paragraphs in order to reduce similarity.  

Please find our responses to their suggestions. Please note that we made the responses to the reviewers in blue color as well as we mentioned the line number.

With many thanks.

The authors

Comments and Suggestions for Authors

This is a well-written case-control research study on macronutrient intake and risks of stomach cancer.  The methods, data analysis and results are clearly written and support the conclusion provided.

Response: Thank you very much for your positive feedback.

Reviewer 2 Report

The authors address a topic of clinical importance. The results are very interesting. The manuscript is well written.

Main concern is the mixture of gastric tumor entities of different etiology and histology. The authors should focus on stomach cancer as gastric adenocarcinoma which is the majority of cases. Otherwise, this should be discussed as a limitation.    

Please describe in detail the selection of controls from the community?

Author Response

Dear Reviewer 2

We would like to thank you and the reviewers for taking the time to carefully review our manuscript and we sincerely appreciate their helpful suggestions to improve the quality of our manuscript. Following the reviewers’ concerns and comments, we made some modifications to the initial version of our manuscript which we marked by track changes. We also rewrite some paragraphs in order to reduce similarity.  

Please find our responses to their suggestions. Please note that we made the responses to the reviewers in blue color as well as we mentioned the line number.

With many thanks.

The authors

Comments and Suggestions for Authors

The authors address a topic of clinical importance. The results are very interesting. The manuscript is well written.

Main concern is the mixture of gastric tumor entities of different etiology and histology. The authors should focus on stomach cancer as gastric adenocarcinoma which is the majority of cases. Otherwise, this should be discussed as a limitation.  

Response:  The majority of enrolled cases (85%) were gastric adenocarcinoma and rest (15%) were gastrointestinal stromal tumor or mucosa associated lymphoid tissue (MALT) lymphomas or carcinoid tumor. The incidence rate of GC in Jordan is low, it is around 2.8 % of all newly diagnosed cancer cases and we attempted to enroll more cases to enlarge the study sample size.  For this reason, we invited all GC patients. It is also impossible to separate GC patients in different groups based on types of GC and examine effect of dietary intake of macronutrients on risk of having different types of GC due to low number of cases diagnosed with gastrointestinal stromal tumor or mucosa associated lymphoid tissue (MALT) lymphomas or carcinoid tumor.  The percentages of gastric adenocarcinoma cases, gastrointestinal stromal tumor, mucosa associated lymphoid tissue (MALT) lymphomas, and carcinoid tumor cases are reported in page 4 Line 110-112. This issue is listed as a limitation in page 11, Line 276-277.

Please describe in detail the selection of controls from the community?

Response:  It has been clarified. The control group (hospital workers, patient visitors, patient escorts, university students and university workers) was recruited conveniently from the community. Individual who did not have a previous history of GC or any type of cancer was selected and frequency matched to cases by age, gender, employment, and marital status. We preferred enrolled control from community rather than hospital because is more accessible and useful to estimate the background prevalence of exposure in the population which gave increase to the cases. However, the selection of controls from a hospital setting increase probability for including individuals with an outcome related to the exposure being studied. In addition, most of hospital-based population suffered from chronic diseases and was following specific diet plan or changing their dietary habits to be more healthier.  Page 4, Line 113-116. Also, there was no clinical confirmation for enrolled controls that were free of GC is mentioned as one of the current study limitations. Page 10, Line 274-275.

Reviewer 3 Report

This is a well-written analysis of a case-control study on diet and gastric cancer risk, which reported significant positive associations for total and several types of dietary fat with risk of gastric cancer.. 

Methods:  Could the authors clarify how controls were selected from the community?  Without this information, it is difficult for readers to evaluate whether selection bias might be present.

It appears that there could be heterogeneity within the gastric cancer patients – by tumor type and possibly by stage (no information is provided on this).  Could the authors provide a bit more information about this within the methods?

With respect to collection of dietary data – What was the timeframe for collection of these data?  It is inferred, based on the description provided in the methods, that the timeframe is one year prior to the interview date for both cases and controls.  If so, for cases, it would include the post-diagnosis timeframe, in which case dietary intake would likely be impacted by the cancer process (symptoms and/or treatment). Given the near 15 kg difference between pre and post diagnosis weight, the authors need to clearly indicate which period the dietary intake data reflect, and adequately discuss this as a limitation (if applicable).

What do pre-diagnosis weight and current weight represent among controls?

The authors present tobacco use (yes/no) in their Table 1, however; models were adjusted for “period of smoking”.  Could table 1 report the data used to adjust models?

Analyses – were tertile cutpoints based on the overall distribution within the study sample (cases and controls combined) or within controls only.  It would also be helpful to report the tertile cutpoints to provide some context to the reader.

Author Response

Dear Reviewer 3:

We would like to thank you and the reviewers for taking the time to carefully review our manuscript and we sincerely appreciate their helpful suggestions to improve the quality of our manuscript. Following the reviewers’ concerns and comments, we made some modifications to the initial version of our manuscript which we marked by track changes. We also rewrite some paragraphs in order to reduce similarity.  

Please find our responses to their suggestions. Please note that we made the responses to the reviewers in blue color as well as we mentioned the line number.

With many thanks.

The authors

Comments and Suggestions for Authors

This is a well-written analysis of a case-control study on diet and gastric cancer risk, which reported significant positive associations for total and several types of dietary fat with risk of gastric cancer.

Methods:  Could the authors clarify how controls were selected from the community?  Without this information, it is difficult for readers to evaluate whether selection bias might be present.

Response:  It has been clarified. The control group (hospital workers, patient visitors, patient escorts, university students and university workers) was recruited conveniently from the community. Individual who did not have a previous history of GC or any type of cancer was selected and frequency matched to cases by age, gender, employment, and marital status. We preferred enrolled control from community rather than hospital because is more accessible and useful to estimate the background prevalence of exposure in the population which gave increase to the cases. However, the selection of controls from a hospital setting increase probability for including individuals with an outcome related to the exposure being studied. In addition, most of hospital-based population suffered from chronic diseases and was following specific diet plan or changing their dietary habits to be more healthier.  Page 4, Line 113-116. Also, there was no clinical confirmation for enrolled controls that were free of GC is mentioned as one of the current study limitations. Page 10, Line 274-275.

It appears that there could be heterogeneity within the gastric cancer patients – by tumor type and possibly by stage (no information is provided on this).  Could the authors provide a bit more information about this within the methods?

Response:  The majority of enrolled cases (85%) were gastric adenocarcinoma and the rest (15%) were gastrointestinal stromal tumor or mucosa associated lymphoid tissue (MALT) lymphomas or carcinoid tumor. The incidence rate of GC in Jordan is low, it is around 2.8 % of all newly diagnosed cancer cases and we attempted to enroll more cases to enlarge the study sample size.  For this reason, we invited all GC patients. It is also impossible to separate GC patients in different groups based on types of GC and examine effect of dietary intake of macronutrients on risk of having different types of GC due to low number of cases diagnosed with gastrointestinal stromal tumor or mucosa associated lymphoid tissue (MALT) lymphomas or carcinoid tumor.  The percentages of gastric adenocarcinoma cases, gastrointestinal stromal tumor, mucosa associated lymphoid tissue (MALT) lymphomas, and carcinoid tumor cases are reported in page 4 Line 110-112. This issue is listed as a limitation on page 11, lines 276-277. The stages of GC have been reported on page 4, Line 113-114.

With respect to the collection of dietary data – What was the timeframe for collection of these data?  It is inferred, based on the description provided in the methods, that the timeframe is one year prior to the interview date for both cases and controls.  If so, for cases, it would include the post-diagnosis timeframe, in which case dietary intake would likely be impacted by the cancer process (symptoms and/or treatment). Given the near 15 kg difference between pre and post diagnosis weight, the authors need to clearly indicate which period the dietary intake data reflect, and adequately discuss this as a limitation (if applicable).

Response: The food frequency questionnaire used in this study assessed dietary intake over the past 12 months. But all of the enrolled cases are newly diagnosed with GC and we excluded participants who being diagnosed with GC for more than six months.  Regarding difference between pre and post diagnosis weight is acceptable because cancer is catabolic condition and cause weight reduction and muscle wasting. Furthermore, signs and symptoms of cancer decrease appetite and amount of eaten foods.  It has been documented that there was recall basis and data collection depends on participants memory. Page: 10, Line 274.

What do pre-diagnosis weight and current weight represent among controls?

Response: It has been changed to be pervious weight. We asked participants to recall their previous weigh.  Page 5, line 139-140 , page 8, line 212, and Page 16, table 1

The authors present tobacco use (yes/no) in their Table 1, however; models were adjusted for “period of smoking”.  Could table 1 report the data used to adjust models?

Response: Result of period of smoking has been inserted in table 1, Page 16.  And page 8, Line 211.

Analyses – were tertile cutpoints based on the overall distribution within the study sample (cases and controls combined) or within controls only.  It would also be helpful to report the tertile cutpoints to provide some context to the reader.

Response: The first and third tertiles are calculated and presented in Table2.  The tertiles for energy and each macronutrient are calculated based on cases alone and controls alone. The intake of energy and macronutrient was different between cases and controls. Page 17.
